# Understanding the Diurnal Oscillation of the Gut Microbiota Using Microbial Culture

**DOI:** 10.3390/life13030831

**Published:** 2023-03-19

**Authors:** Guilherme Amando, André Tonon, Débora Constantino, Maria Paz Hidalgo, Pabulo Henrique Rampelotto, Francisco Montagner

**Affiliations:** 1Chronobiology and Sleep Laboratory, Hospital de Clínicas de Porto Alegre (HCPA), Porto Alegre 90410-000, Brazil; 2Graduate Program in Psychiatry and Behavioral Sciences, Faculty of Medical Sciences, Federal University of Rio Grande do Sul (UFRGS), Porto Alegre 90410-000, Brazil; 3Faculty of Health and Medical Sciences, University of Surrey, Guildford GU2 7XH, UK; 4Department of Psychiatry and Legal Medicine, Faculty of Medical Sciences, Federal University of Rio Grande do Sul (UFRGS), Porto Alegre 90410-000, Brazil; 5Graduate Program in Pharmacology and Therapeutics, Institute of Health Sciences, Federal University of Rio Grande do Sul (UFRGS), Porto Alegre 90410-000, Brazil; 6Graduate Program in Dentistry, Faculty of Dental Sciences, Federal University of Rio Grande do Sul (UFRGS), Porto Alegre 90410-000, Brazil

**Keywords:** microbiology, light–dark cycle, circadian rhythm, anaerobic bacteria, fungi

## Abstract

The composition of the gut microbiota oscillates according to the light–dark cycle. However, the existing literature demonstrates these oscillations only by molecular methods. Microbial cultures are an interesting method for studying metabolically active microorganisms. In this work, we aimed to understand the diurnal oscillation of the intestinal microbiota in Wistar male rats through microbial culture analysis. Over a 24 h period, three animals were euthanized every 6 h. Intestinal segments were dissected immediately after euthanasia and diluted in phosphate-buffered saline (PBS) for plating in different culture media. The CFU/mL counts in feces samples cultured in the Brucella medium were significantly higher at ZT0, followed by ZT6, ZT18, and ZT12 (*p* = 0.0156), which demonstrated the diurnal oscillation of metabolically active anaerobic bacteria every 6 h using microbial culture. In addition, quantitative differences were demonstrated in anaerobic bacteria and fungi in different gastrointestinal tract tissues.

## 1. Introduction

The circadian system provides rhythmicity to several physiological processes of living beings (e.g., hormonal secretion, temperature, and blood pressure) and is synchronized to cyclic environmental signals (i.e., external clues of temporality) in a process known as entrainment. Daily rhythms of approximately 24 h are called circadian rhythms [1,2,3]. The entrainment of the circadian system (endogenous) with environmental signals (exogenous) is crucial for maintaining health [4,5] and occurs mainly by light, the main zeitgeber (“time giver” in German; environmental cue of the passage of time) [6,7]. Moreover, circadian misalignment (mismatch between internal and external rhythms) is often observed to be associated with metabolic and gastrointestinal disorders [8,9,10,11].

Recent studies have demonstrated that the intestinal microbiota also undergoes diurnal compositional and functional oscillations, which are driven by the light–dark cycle [12,13,14]. This oscillation has been demonstrated in adults (and associated with short-chain fatty acid levels) [15] and in mice kept under different light exposure protocols, where the group in a forced jet lag protocol showed arrhythmicity of these populations [12]. Diurnal microbial pattern drives, in turn, the host physiology and metabolism in many ways [16]. Additionally, diabetes type 2 was associated with disrupted rhythmicity of microbial populations in the gut of a longitudinal population-based cohort [17]. Similarly, another study suggested that this microenvironment could be an independent contributor to elevated serum amino acid levels in participants with insulin resistance [18]. In addition to its diurnal oscillation, gut microbiota is influenced by biogeography, allowing the establishment of different microbial populations in each intestinal microenvironment [19,20].

Assessing microbial populations at different times is crucial for understanding such diurnal oscillations. Different studies have showed that pathological outcomes are associated with circadian rhythm disruption [21,22], which also induces microbiota dysbiosis [9,23]. Furthermore, a recent study demonstrated the oscillation of different microbial phyla in healthy mice, but also the lack of rhythmicity of mice in constant conditions (e.g., 24 h of constant darkness every day) and knockout for clock genes [24]. Thus, it is important to address multiple time points to better characterize any gut microbiota change.

Different approaches can be used to characterize the gut microbiota. With the advent of high-throughput sequencing, most studies nowadays characterize these microbial communities by metagenomics [25,26]. However, metagenomics does not discriminate viable from non-viable cells as well as active or quiescent microorganisms. This may be mitigated using RNA sequencing, but metatranscriptomics is still a high-cost technology inaccessible to most researchers. The microbial culture method can be used to study the metabolically active populations that represent major components of the intestinal microbiota, such as cultivable bacteria and fungi [27,28]. In addition, this traditional method can be the first step toward culturomics [27,29], where new microbial species can be taxonomically validated and officially named. Moreover, the microbial culture could complement the microbiota characterization provided by metagenomics because this molecular method may also detect inviable microorganisms, giving a false assumption about their role [30]. To date, a few studies have evaluated the diurnal oscillations in intestinal microbial populations, but none have used the microbial culture method.

To fulfill this need, the aim of this study was to analyze the microbial cultures of different tissues of the gastrointestinal tract (i.e., cecum and rectum) as well as feces taken directly from the rectal ampoule at different zeitgeber times to understand the diurnal oscillation of the gut microbiota of male Wistar rats.

## 2. Material and Methods

### 2.1. Animals

Male Wistar rats (*n* = 12) at 13 weeks of age were obtained from the Centro de Reprodução e Experimentação em Animais de Laboratório (CREAL, Porto Alegre, Brazil). The sample size calculation was based on work by Thaiss et al. (2016) and was defined based on the evaluation of circadian variation in the composition of the gut microbiota [13]. Before the protocol started, the rats had been moved daily among cages to ensure that all animals were housed with all other animals for at least 2 days. This procedure enables the induction of a uniform baseline microbiota across all of them. All animals were fed ad libitum with a regular chow diet and kept under the same controlled conditions of temperature (22 ± 2 °C), humidity, reduced noise exposure, and a standard photoperiod (12 h light and 12 h dark, with lights on at 7:00 am). These conditions are the standard for a bioterium of rats that is not intended to cause any intervention or stress on the animals. The research protocol was approved by the Institutional Animal Research Ethics Committee at the Hospital de Clínicas de Porto Alegre (CEUA/HCPA, protocol number 2019–0413), following the recommendations set out in the ARRIVE guidelines [31].

### 2.2. Experimental Protocol

Over the course of a 24 h period, 3 animals were euthanized every 6 h. Immediately after euthanasia, the intestines were dissected on a sterile field. The small intestine was separated and sectioned to obtain samples of cecum and rectum. Feces were taken directly from the last portion of rectal ampoule. The fragments of each sample (i.e., cecum, rectum, and feces) were weighed and stored per animal in separate Falcon tubes, preparatory to microbial culturing. After collecting the sample, sterilized phosphate-buffered saline (PBS) solution was added to the tube, maintaining the proportion of 1 mg of sample to 1 μL of PBS.

The procedures for the use of scientific animals were also conducted in accordance with the Guide for the Care and Use of Laboratory Animals [32]. This study was registered with the National System of Genetic Resource Management and Associated Traditional Knowledge (SisGen), under registration number ABFDC4F.

### 2.3. Serial Dilutions and Culture Media

Serial dilutions were performed starting from the initial sample to 1/10, 1/1000, and 1/100,000 using PBS solution. The undiluted sample was pipetted at the center of a Petri dish. Each Petri dish (90 × 15 mm) was divided into three equally distributed sections. Three 25 μL drops of each dilution were pipetted into one of the three sections, one section per dilution, with no contact between drops [33]. The periods, conditions of incubation, and purpose of each culture medium used are described in Figure 1. The culture media (purchased from HiMedia^®^, Mumbai, India) chosen were selected to cover the main groups of cultivable microorganisms from the gut. This method was used only for the purpose of counting these microorganisms.

The number of colony-forming units (CFU) present in a single drop (25 μL per drop) was determined for each culture media by multiplying the number of CFU observed during counting by 40, 400, 40,000, or 4,000,000, depending on the dilution being counted, as obtained by the following equation: CFU initial sample/mL (total) = CFU count × 40 × dilution factor. These dilutions were chosen to allow better visualization of the cultivated colonies.

In cases where an extremely high number of colonies made counting of CFU impossible, it was decided to use the maximum number possible of CFU in a 25 μL drop (i.e., 75 CFU) at dilution of 10^−5^, as described by Naghili et al. (2013) [33]. Petri plates with patterns indicative of contamination were excluded from analyses.

### 2.4. Time Measurements

The exact time referring to the light and dark phase is of major importance to our study because it is directly linked to the outcome. Thus, the results were described through the measure of time called zeitgeber time (ZT), commonly used to measure time in chronobiological studies. The ZTs correspond to six-hour intervals within a 24 h period, starting at the beginning of the light phase. The experimental protocol started at ZT0, which corresponded to the moment when the lights in the bioterium were turned on, namely 7 am for the researchers.

### 2.5. Statistical Analysis

All analyses were performed in GraphPad Prism version 8.4.2, with significance at *p* < 0.05. Study variables are described as median (interquartile ranges) because there was no parametric distribution of the data (see Appendix A). For comparisons of CFU/mL counts between different sites and ZTs, the Kruskal–Wallis test followed by Dunn multiple comparison was used.

## 3. Results

The CFU/mL results for each culture medium for samples collected at each ZT are expressed as their respective medians (see Figure 2 or Appendix A).

### 3.1. Differences between ZTs

In samples plated in BA medium, the CFU/mL count in feces was significantly higher (*p* = 0.0156) at ZT0, followed by ZT6, ZT18, and ZT12. This indicates that the microorganisms cultivated in this medium peaked their concentration in the beginning of the light phase (ZT0), decreased in the middle of the light phase (ZT6), reached the lowest point at the end of light phase (ZT12), and started to increase their concentration again in the middle of the dark phase (ZT18). There was a trend towards statistical significance (*p* = 0.0662) for the RT samples in the MS medium, with the highest CFU/mL count observed at ZT6, followed by ZT18, ZT12, and ZT0. Similarly, as observed in BA medium, this trend indicates that higher concentrations of microorganisms were observed during the light phase (ZT0 and ZT6 = light phase). Specifically, the concentration of CFU/mL peaked in the middle of the light phase (ZT6), reached the lowest point at the end of the light phase (ZT12), increased its concentration again in the middle of the dark phase (ZT18), and decreased again in the beginning of the light phase (ZT0). There was no statistical difference between ZTs in BHI and SA media.

### 3.2. Biogeographic Differences

In samples plated in BHI medium, the CFU/mL count was higher in FC than in CE and higher in CE than in RT at both ZT6 (*p* = 0.0321) and at ZT12 (*p* = 0.0036). This indicates that there is a different concentration of microorganisms and could indicate a distinct diversity in the middle of the light phase (ZT6) when comparing sample sites. There was a trend towards statistical significance at ZT18 (*p* = 0.075) following the same pattern (i.e., FC > CE > RT) (Figure 2), indicating the same difference observed in BHI medium, but in the middle of the dark phase (ZT18). In samples plated in SA and MS media, there were statistically significant differences at ZT0 (*p* = 0.0214; *p* = 0.0107) following the same pattern, also indicating the same biogeographic differences observed in other media, but at the beginning of the light phase (ZT0). There was no statistical difference in samples plated in BA medium.

## 4. Discussion

In this preliminary work, we present evidence of the diurnal oscillation of microbial populations in the gut and differences in their composition using microbial culture methods. Our results also indicated differences among sampling sites, demonstrating biogeographical differences in different GI tract sites. Our main findings showed that the feces samples cultivated in Brucella agar (i.e., facultative anaerobic bacteria) exhibited microbial variation between ZTs, with higher concentrations of CFU/mL at the beginning and in the middle of the light period (i.e., resting phase). Using a metagenomic approach, Thaiss et al. (2014) found evidence of rhythmic disruption in microbial populations of Ruminococcaceae, a family belonging to Firmicutes (predominantly anaerobic) [12]. Two other species from this phylum also exhibited changes in relative abundance over the course of the day. These species had a higher relative abundance value in the resting phase [12]. Here, our results suggested a similar oscillation of bacterial concentration, although we used a microbial culture method, collecting species from different sites.

In this study, the laboratory growth conditions favored facultative anaerobic bacteria of the genus Streptococcus to be cultivated in the BA medium. There was a high number of microorganisms at the beginning of the light phase, observed by the peak in CFU/mL counting at ZT0. Li et al. (2017) observed that the gastrointestinal microbiome of rats is predominantly composed of the phylum Firmicutes, regardless of the tissue collected [20]. Similarly, Thaiss et al. (2016) demonstrated that *Mucispirillum schaedler*, a species from the Deferribacterota phylum that is also anaerobic, had higher concentrations at the beginning of the light phase [13]. Both studies seem to agree with the results of this study.

Our findings also revealed differences between collection sites in CFU counts during the middle of the light phase (ZT6), and afterward at the end of the light phase (ZT12) in BHI media (facultative anaerobic bacteria). These differences indicated that feces exhibited higher concentrations of microorganisms than the cecum and rectum, respectively. Relative abundance regarding the rhythmicity of bacterial communities in different sample sites of the intestinal tract of rats has also been reported throughout the day [12,13,14]. The control group exhibited a different configuration of bacterial genera depending on the time of day. However, rats submitted to a chronodisruption protocol lost the oscillatory pattern [12]. Another study observed similar rhythmicity of several microbial taxa once every 4 h [34]. We observed different CFU counts depending on the time of day for bacteria cultivated in the BHI media, representing highly abundant and metabolically active microorganisms. 

Results from the Mitis Salivarius and BHI culture media indicate biogeographical interference due to the highest CFU/mL values for FC, followed by CE and RT. Recently, the bacterial communities present in each segment of the gastrointestinal tract (feces and the contents of the large and small intestine) of male rats were described using metagenomics [35]. Higher bacterial diversity and proportion for bacterial components of different genera and families in the large intestine, but mainly in feces, were observed. These genera and families are mostly represented by anaerobic bacteria. Furthermore, Li et al. (2017) also presented results consistent with the study mentioned before. The diversity found was attributed to a more complex micro-ecosystem in the large intestine of rats, resulting in higher bacterial concentrations in feces [20]. Although cultures do not enable all microorganisms to be cultivated separately, our results from Mitis Salivarius and BHI are consistent with the studies mentioned above. 

Fungi are an important component of the gut microbiome, but they are frequently neglected in most studies [36]. Here, we observed a higher CFU/mL count at the beginning of the light phase (i.e., rest period). Chen et al. (2018) demonstrated that *Aspergillus fumigatus* colonization in rats knocked out for different clock genes differed depending on the time at which the animal was infected. Moreover, there was a difference in CFU counts between the times, with the ZT0 having the highest number of CFU counts in the lung compared to ZT12. The authors suggested that the interaction between the host and this particular fungus may be under some circadian control [37]. In our study, higher fungal counts were also observed at the beginning of the morning (ZT0) in feces when compared to the other collection sites. However, *A. fumigatus* cannot be cultured in the SA medium. In our study, we incubated the SA media plates at room temperature (25 °C ± 2 °C) following incubation at 37 °C (see Figure 1). As described by Hazen and Hazen (1987), *Candida albicans* room-temperature-grown cells were generally less sensitive to environmental perturbation and germinated more uniformly than cells grown at 37 °C [38]. Furthermore, a different study suggested that there is a synergy between this fungus and species of Bacteroides. The authors observed that while Bacteroides’ growth was significantly enhanced in co-culture with *C. albicans*, the cell concentrations of some strains of *C. albicans* were unaffected by the presence of specific Bacteroides species. This result suggests the cells of *C. albicans* may serve as an additional nutrient source for the bacteria in anaerobic regions of the gut [39]. Here, this synergy could explain the higher concentrations of CFU/mL in feces when compared to other collection sites. Bacteroides species are mainly represented in BA media and, as described before, the concentration of CFU/mL in feces peaked at ZT0. This peak may have influenced the concentration of CFU/mL of fungi evaluated in the SA culture medium. Furthermore, this influence could have led to differentiation in the number of microorganisms between sites, resulting in a higher concentration of fungi in feces than in other sites.

To date, this is the first study to highlight the diurnal oscillation and biogeographical differences of the gut microbiota using culture media, and the results presented here have relevant implications. First, the identification of a diurnal oscillation of metabolically active anaerobic bacteria once every 6 h indicates that this component possibly impacts most studies involving the gut microbiota. Therefore, the evaluation of this oscillation is of major importance to ensure the reproducibility and reliability of future research. Furthermore, it is important to emphasize how different microorganisms can be in one day. For instance, in the same direction of melatonin’s phase response, it is extremely important that every study focused on evaluating microbial communities should aim to assess at least three different moments during a day. Thus, future studies should consider a chronobiological design for the collection and evaluation of the outcome of interest, regardless of the microbiota. Moreover, our study provides a model of what to expect from regular variations of intestinal microorganisms because the rats were kept in normal conditions in every aspect (i.e., food intake, light exposure, no stressors). It is important to note that our preliminary results were obtained over the course of one 24 h period. Additional studies should be performed with longer periods of time to confirm the periodicity of our findings, but also to compare with different photoperiods. It is important to understand the extension of how metabolically active microorganisms behave in constant photoperiods (i.e., 24 h of constant light or darkness) to elucidate the role of light and its influence on the gut microbiota. It is also important to note that our small sample size was based on a sample size calculation, which supports the fact that our procedure and results derived from our methodology are not random. Similarly, there are plenty of studies evaluating outcomes related to the rhythmicity of the gut microbiota using small samples similar to ours [12,13]. In addition, these studies usually use some type of intervention, whether light (e.g., constant darkness) or medication (usually antibiotics). Thus, future studies should aim to increase the protocol time to assess whether the rhythmicity observed in some culture media is maintained over longer periods.

Second, the use of different culture media enables the detection of changes in intestinal microbial compositions at different collection sites, providing the baseline for the application of more advanced culture methods in future studies analyzing the circadian rhythm of the gut microbiota. Thus, this traditional method can be the first step toward the use of culturomics in circadian rhythm research, where new microbial species could be taxonomically validated and officially named. Moreover, it is important to note that one of our goals here was to observe the diurnal oscillation of anaerobic microorganisms more carefully because they also constitute a large portion of the gut microbiota. Furthermore, we also aimed to observe these oscillations in fungi, which are also an important component of the gut microbiome, though they are frequently neglected in most studies. Hence, we chose culture media that could fulfill these goals. More specifically, BA and MS culture media were chosen to cover most anaerobic bacteria, whereas BA and SA would cover aerobic microorganisms (bacteria and fungi, respectively; see Figure 1). Additionally, there is a portion of the gut microbiota that comprises non-culturable microorganisms and that plays an extremely important role in the physiology and homeostasis of this microenvironment. To meet this need, next generation sequencing techniques, such as amplicon sequencing (e.g., 16S rRNA), could be used to address non-culturable taxa characterization. Currently, this technique is the most widely applied in microbiome studies [25,26] and has plenty of standardized analytical pipelines aiming to produce accurate and reproducible results, thereby allowing comparison between studies [40]. It is important to note that we only used male rats due to uncertainty regarding whether there is any interaction with the estrous cycle and microbial communities in the gut. The literature has shown that the estrous cycle interferes with several physiological processes in rats [41,42,43]. We do not know for sure whether this process interferes with the rhythmicity of the gut microbiota, but to avoid the risk of influencing this microenvironment, we chose not to use female rats. Future studies should evaluate whether there is some interaction between the estrous cycle and microbial communities while controlling the phases of the estrous cycle that occur after the vaginal opening.

Third, we chose to use non-specific culture media in conditions to favor the growth of strict and facultative anaerobes, allowing the growth of a large group of cultivable microorganisms. Therefore, it is expected that they vastly cover the expected populations in the gut microbiota. It would also be interesting to evaluate the oscillation of specific targets, such as *Klebsiella* spp., which is fundamentally important in studies of hospital-acquired infections and antibiotic resistance [44,45], as well as *Escherichia coli*, which has been observed to be associated with several intestinal diseases and virulence potential [46,47,48]. Furthermore, future studies should focus on evaluating lactic acid bacteria due to their relevance as one of the major components of microorganisms in the gut microbiota. Furthermore, plenty of culture media could be used to grow these microorganisms, such as MRS (De Man, Rogosa, and Sharpe) agar, for example. Other physiological and molecular aspects may be associated with these diurnal oscillations in the gut microbiota and should be observed in future studies. Wang et al. (2021) aimed to explore the effect of different feeding patterns on intestinal health through, among other parameters, the expression of short-chain fatty acids (SCFA), intestinal tight junction proteins, clock genes, and the diurnal rhythm of microbial populations in rabbits [49]. At the beginning of the dark phase (ZT13), levels of butyric acid (SCFA) were higher in the control group when compared to the restricted food group (intervention group). However, in the same ZT, levels of CLAUDIN-1 (intestinal tight junction proteins) and PER1 (clock gene) were significantly higher in the intervention group. In addition, there were different percentages of relative abundance in Firmicutes and Bacteroidetes phyla in ZT13; the first were cited higher in the intervention group but the second were cited lower in the same group [49]. Here, we observed different CFU/mL counting at ZT12 among the culture media, and the BA medium had the lowest count. This suggests a similar behavior observed of the Bacteroidetes phylum’s relative abundance in the control group. Anaerobic bacteria are also present in this phylum. Therefore, the results of Wang et al. (2021) seem to agree with our results. Furthermore, aiming to fully understand the mechanisms that underlie the diurnal rhythmicity of the gut microbiota, future studies should aim to evaluate specific molecules that underlie the physiology of the gastrointestinal tract. 

Lastly, it is important to emphasize that every methodology of assessing microbial populations has both advantages and limitations. The microbial culture-based method is time-consuming, dependent on culture media and incubation characteristics, and unsuitable for fastidious bacteria growth with complex nutritional requirements. Furthermore, as stated before, some microorganisms that cannot be cultivable and are crucial to understanding gut microbiota complexity. Remarkably, advances in molecular approaches contribute to several topics of microbiology research and have brought about a significant body of new knowledge regarding not only diseases, but also health aspects. However, as stated before, most molecular methods are not able to fully demonstrate the aspects of several microbiotas. The characteristics of both culture-based and molecular methodologies were previously elucidated by Siqueira and Rôças (2005) [30]. The authors suggest a workflow on how to combine both methodologies to achieve a better understanding of the microenvironment landscape. Hence, we reinforce the idea that future studies should aim to incorporate both molecular and culture-based methods for a better understanding of the microbiota of interest.

## Figures and Tables

**Figure 1 life-13-00831-f001:**
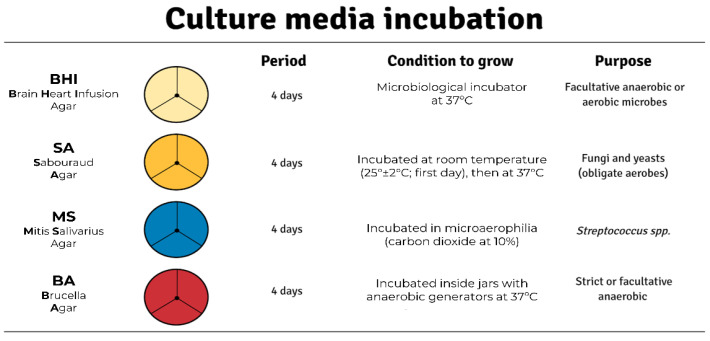
Periods, conditions of incubation, and purpose of each culture medium. The three divided sections in the dish represented one section per dilution (i.e., 1/10, 1/1000, and 1/100,000).

**Figure 2 life-13-00831-f002:**
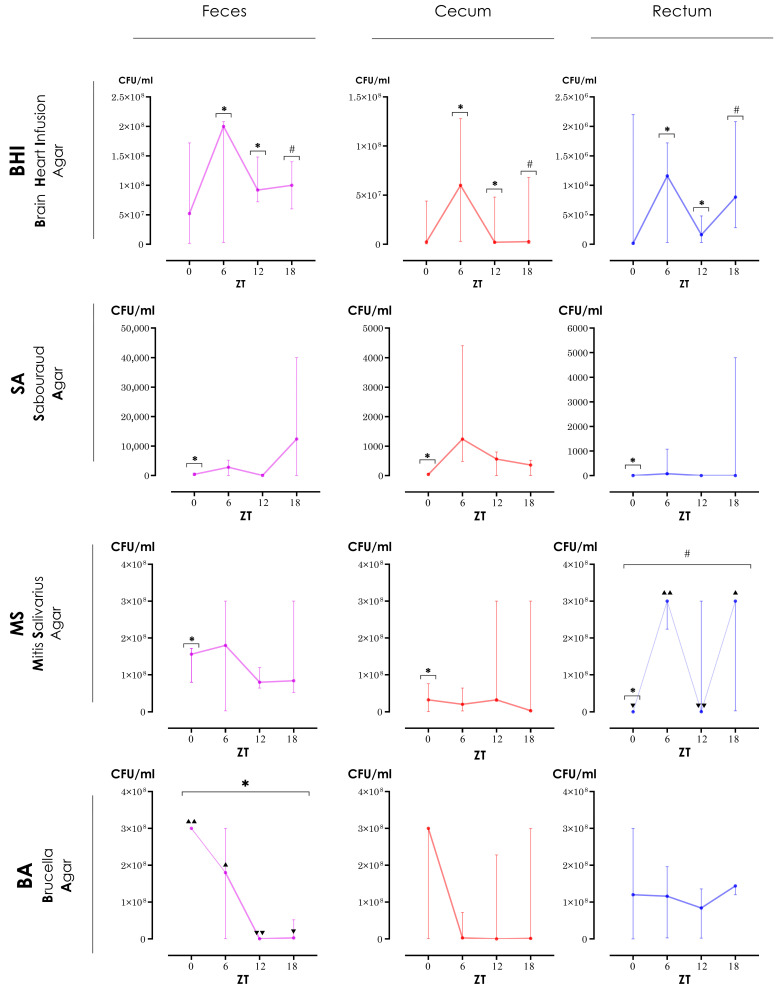
Panel of scattered dot plots illustrating median (IQR) of CFU/mL counts at each ZT. Each line corresponds to a culture medium, while each column corresponds to a collection site. Kruskal–Wallis test followed by Dunn’s test for multiple comparisons were performed; Statistical significances (*p* < 0.05) are marked with “*”; Trends to significance (*p* > 0.05 but < 0.07) are marked with “#”; Differences in CFU/mL counts between ZTs are marked with black triangles. The direction and the number of black triangles indicate the hierarchical position of each point based on the CFU/mL count. ▲▲ = highest CFU/mL counting; ▲ = second highest CFU/mL counting; ▼▼ = lowest CFU/mL counting; ▼ = second lowest CFU/mL counting.

## Data Availability

Not applicable.

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
