# Peer review of "Understanding the Diurnal Oscillation of the Gut Microbiota Using Microbial Culture"

_life, 2023, doi:10.3390/life13030831_

Round 1

Reviewer 1 Report (New Reviewer)

Author Response

See attached.

Reviewer 2 Report (New Reviewer)

Major revisions:

Introduction: The first paragraph does a poor job with several circadian definitions. Most importantly, Circadian rhythms are not simply 24hr rhythms. (1) They must be endogenous and persist in constant conditions and (2) They must be able to be entrained to the environment. Any review by Joe Takahashi should provide a good source for reference. You use the term Zietgeber within the later manuscript, here you can define the term. Rhythm disruptions are often called chronodisruption or circadian misalignment.

Methods/Results: As you are using non-parametric statistics, it would be good to describe the fact and which assumptions were violated within your data. The reviewer worries that the sample size is so small, you need to show the errors within your figures. The legend of figure 2 is not clear and hard to understand, the authors should improve this for clarity.  

Discussion: The final three paragraphs are clearly setup as conclusion points, however, the ideas within each is not clearly outline and should be improved to communicate their points. An example is in the second to last paragraph, this brings up future circadian experiments; was this an attempt to outline a clear limitation of the study as not a circadian design? The authors should make this more clear, animals were in a LD cycle and so these are diurnal rhythms, to determine if circadian rhythms then the experiment should be replicated in constant dark conditions with enough time to prevent after effects. Finally, the manuscript ends awkwardly, the authors should consider having a sentence or two that summaries the importance of the papers findings and a concise future direction/impact on the field. 

Minor revisions:

Page 7 a sentence begins with "Worth of note," did the authors mean worthy or notice?

Author Response

See attached.

Reviewer 3 Report (New Reviewer)

The manuscript of Guilherme Amando and colleagues presents a very interesting topic in the field of the circadian rhythms involved in the temporal homeostasis of animals. However, the communication is riddled with mistakes and there are several points that the authors must clarify and extensively improve to consider their work for publication. See all my suggestions in the following detailed report.

Author Response

See attached.

Reviewer 4 Report (New Reviewer)

Recommendation:

Major Revision 

Title: Understanding the Diurnal Oscillation of the Gut Microbiota Using Microbial Culture

Major Comments: 

1) The study suggests to integrate culture-based techniques in assessing diurnal oscillation of the gut microbiota. The author should further discuss how different it is from previous studies done in mammalian gut microbiota. Are there evidences (published papers) that metabarcoding and metagenome analysis are not sufficient enough to describe the processes involve in gut microbiota. May I request the author(s) to provide a literature survey that will further support and clarify this concern. The disadvantages of culture-based techniques should also be added in manuscript. 

2) It seems like that the authors used a limited number of culture medium (only four) in this study. The author should explain their basis of choosing the culture media. Are those already optimized culture media that will give a good representation of the overall diversity in the gut microbiota or not? Lactic acid bacteria is considered one of the major organisms present in the gut microbiome and grows well in MRS (De Man Rogosa and Sharpe) Agar. Why not consider this medium in this study? I think additional culture medium should be included in the experiment. 

3) The authors should also include molecular data (metagenome and metabarcoding) to serve as a control in their experiments. Comparison of the trend in the relative abundance between molecular and culture based techniques is important to prove that the data in both methods support each other. The author should also address this concern in the main manuscript.

4) Recently, there are reports that both the host circadian system and mammalian host gender influence the rhythmicity of the total bacterial load and taxonomic abundances in the fecal microbiota. The author(s) should also include these factors in the design of experiment to further support the claim to include culture based techniques in assessing diurnal oscillation of the gut microbiota. The authors should include discussions on the effect circadian factors in microbiome studies especially in analyzing absolute abundance in understanding the microbiota.

5) Recent studies suggests that multiple immune- related molecules have been linked to the diurnal microbiota-host interaction. Molecules such as Reg3γ, IgA, and MHCII are secreted or expressed on the gut surface and directly interact with intestinal bacteria. In addition, these molecules are also strongly influenced by dietary patterns (eg. time-restricted feeding and high-fat diet), which are already known to modulate microbial rhythms and peripheral clocks. I suggests that the authors should relate the results of their experiments to these important effector molecules and include it in the discussion part of the manuscript. 

Author Response

Reviewer #4 round 1

We thank the reviewer for the additional comments and suggestions. Below we provide the point-by-point reply to them all.

Major Comments:

1) The study suggests to integrate culture-based techniques in assessing diurnal oscillation of the gut microbiota. The author should further discuss how different it is from previous studies done in mammalian gut microbiota. Are there evidences (published papers) that metabarcoding and metagenome analysis are not sufficient enough to describe the processes involve in gut microbiota. May I request the author(s) to provide a literature survey that will further support and clarify this concern. The disadvantages of culture-based techniques should also be added in manuscript.

Reply: As stated in the introduction (line 75), we understand that the molecular methods that are widely used in the literature (e.g., metagenomics) cannot differentiate viable from non-viable cells as well as active or quiescent microorganisms. In the same direction, the literature does not have a clear consensus as to what extent metabarcoding is quantitative. Furthermore, some studies suggest caution regarding the use of metabarcoding until this methodology is optimized for accuracy and precision. Therefore, one of the main points of the discussion was to emphasize that microbial culture should be used together with molecular methods to better characterize the gut microbiota. For this reason, we have added to the discussion the following sentences:

(line 384): “Lastly, it is important to emphasize that every methodology for assessing microbial populations has advantages and limitations. The microbial culture-based method is time-consuming, dependent on culture media and incubation characteristics, and unsuitable  for fastidious bacteria growth with complex nutritional requirements. Also, as stated before, some microorganisms that cannot be cultivable and are crucial to understanding the gut microbiota complexity.”

2) It seems like that the authors used a limited number of culture medium (only four) in this study. The author should explain their basis of choosing the culture media. Are those already optimized culture media that will give a good representation of the overall diversity in the gut microbiota or not? Lactic acid bacteria is considered one of the major organisms present in the gut microbiome and grows well in MRS (De Man Rogosa and Sharpe) Agar. Why not consider this medium in this study? I think additional culture medium should be included in the experiment.

Reply: We understand the reviewer’s concern about the chosen culture media and we address this concern in the discussion. Also, we agree that adding more media to the experiment would be very interesting.  However, it is important to note that one of our goals here was to observe the diurnal oscillation of anaerobic microorganisms more carefully, since they are also a large portion of the gut microbiota. Furthermore, we also aimed to observe these oscillations in fungi, which is also a critical component of the gut microbiome, but they are frequently neglected in most studies. As stated in the manuscript, fungi seem to interact with anaerobic microorganisms, but mainly bacteria. Therefore, this interaction would be interesting to observe in our experiment. Thus, we chose culture media that could fulfill these goals (i.e., BHI, BA, SA, and MS culture media) and that we were used to working with in previous studies. You can see below the changes in red or directly in the manuscript.

(line 332): “Moreover, it is important to note that one of our goals here was to observe the diurnal oscillation of anaerobic microorganisms more carefully , since they are also a large portion of the gut microbiota. Furthermore, we also aimed to observe these oscillations in fungi, which is also an important component of the gut microbiome, but they are frequently neglected in most studies. Hence, we choose culture media that could fulfill these goals. More specifically, BA and MS culture media were chosen to cover most anaerobic bacteria, whereas BA and SA would cover aerobic microorganisms (bacteria and fungi, respectively; see Figure 1).”

(line 358):It would also be interesting to evaluate the oscillation of specific targets, such as Klebsiella spp, which is fundamentally important in studies of hospital-acquired infections and antibiotic resistance [41, 42] as well as Escherichia coli, which have been observed to be associated with several intestinal diseases and virulence potential [43–45]. Furthermore, future studies should focus on evaluating lactic acid bacteria due to their relevance as one of the major components of microorganisms in the gut microbiota. Also, plenty of culture media could be used to grow these microorganisms, such as MRS (De Man, Rogosa & Sharpe) Agar, for example.”

3) The authors should also include molecular data (metagenome and metabarcoding) to serve as a control in their experiments. Comparison of the trend in the relative abundance between molecular and culture based techniques is important to prove that the data in both methods support each other. The author should also address this concern in the main manuscript.

Reply: We agree that it would be interesting to compare both molecular and culture-based techniques. However, the results presented here are part of preliminary work. As discussed in the manuscript, this is the first study to highlight the gut microbiota’s diurnal oscillation and biogeographical differences using culture media. Therefore, we aimed to achieve a controlled and interference-free protocol in order to observe these microorganisms’ diurnal oscillation during a homeostatic period. This way, our findings could help future studies to implement this methodology and what to expect in a control group. As stated in the discussion, Siqueira and Roças (2005) clarify that culture-based and molecular methods could be complementary, mainly due to how the methods evaluate the number of microorganisms. In the discussion, we highlighted that microbial culture can observe metabolically active microorganisms. Therefore, it could be an alternative to complement intestinal microbiota analysis. In the introduction, we also mentioned that metagenomics does not discriminate viable from non-viable cells or active or quiescent microorganisms. It may be mitigated by using RNA sequencing, but metatranscriptomics is still a high-cost technology, inaccessible to most researchers.

4) Recently, there are reports that both the host circadian system and mammalian host gender influence the rhythmicity of the total bacterial load and taxonomic abundances in the fecal microbiota. The author(s) should also include these factors in the design of experiment to further support the claim to include culture based techniques in assessing diurnal oscillation of the gut microbiota. The authors should include discussions on the effect circadian factors in microbiome studies especially in analyzing absolute abundance in understanding the microbiota.

Reply: We agree that it would be interesting to evaluate the rhythmicity of the gut microbiota in a female sample as well. Indeed, gender may be an important influence factor in the gut microbiota's rhythmicity. However, gender is also an important factor in chronobiology studies. In animal models, the estrous cycle phases (equivalent to the menstrual cycle) seem to influence physiological aspects of these animals, from hormonal regulation to behavioral changes. In our last literature search, we did not find any study that observed changes in the gut microbiota in different phases of the estrous cycle. Therefore, it is still uncertain whether there is any interaction between the estrous cycle and microbial communities in the gut. Moreover, our sampling was adequate for this preliminary pilot study. First, we performed a sample size calculation before starting any procedure to ensure that the number of animals would be sufficient to carry out this experiment. Therefore, we chose to use 12 animals after this calculation, since that would be the minimum number necessary to obtain the expected result. Similarly, plenty of studies evaluating outcomes related to the rhythmicity of the gut microbiota using small samples similar to ours . In addition, these studies usually use some intervention, whether light (e.g., constant darkness) or medication (usually antibiotics). Thus, future studies should aim to increase the protocol time to assess whether the rhythmicity observed in some culture media is maintained over more extended periods. We think that this discussion is important to address, so we have added the following sentences to our manuscript:

(method section, line 115): “Male Wistar rats (n = 12) at 13 weeks of age were obtained from the Centro de Reprodução e Experimentação em Animais de Laboratório (CREAL, Porto Alegre, Brazil). The sample size calculation was based on Thaiss et al (2016) and was defined based on the evaluation of circadian variation in the composition of the gut microbiota [13].”

(line 318):Also, it is important to note that our small sample size was based on a sample size calculation. which supports the fact that our procedure and results derived from our methodology are not random. Similarly, there are plenty of studies evaluating outcomes related to the rhythmicity of the gut microbiota using small samples similar to ours (REFs). In addition, these studies usually use some type of intervention, whether light (e.g., constant darkness) or medication (usually antibiotics). Thus, future studies should aim to increase the protocol time to assess whether the rhythmicity observed in some culture media is maintained over longer periods.”

(line 346): Also, it is important to notice that we only used male rats due to the uncertainty regarding whether there is any interaction with the estrous cycle and microbial communities in the gut. The literature has shown that the estrous cycle interferes with several physiological processes in rats (REFs). We do not know for sure whether this process interferes with the rhythmicity of the gut microbiota, but to avoid the risk of influencing this microenvironment, we chose not to use them. Future studies should evaluate whether there is some interaction between the estrous cycle and microbial communities, but control the phases of the estrous cycle that occur after the vaginal opening.

5) Recent studies suggests that multiple immune- related molecules have been linked to the diurnal microbiota-host interaction. Molecules such as Reg3γ, IgA, and MHCII are secreted or expressed on the gut surface and directly interact with intestinal bacteria. In addition, these molecules are also strongly influenced by dietary patterns (eg. time-restricted feeding and high-fat diet), which are already known to modulate microbial rhythms and peripheral clocks. I suggests that the authors should relate the results of their experiments to these important effector molecules and include it in the discussion part of the manuscript.

Reply: We appreciate the reviewer's comment and agree that specific molecules interact with intestinal bacteria and modulate peripheral clocks' expression. Future studies should evaluate specific molecules that underlie the physiology of the gastrointestinal tract. We think that this discussion is important to address, so we have added the following sentences to our manuscript:

(line 365): Other physiological and molecular aspects may be associated with these diurnal oscillations in the gut microbiota and should be observed in future studies. Wang and colleagues (2021) aimed to explore the effect of different feeding patterns on intestinal health through, among other parameters, the expression of short-chain fatty acids (SCFA), intestinal tight junction proteins, clock genes, and the diurnal rhythm of microbial populations in rabbits. At the beginning of the dark phase (ZT13), levels of butyric acid (SCFA) were higher in the control group when compared to the restricted food group (intervention group). However, in the same ZT, levels of CLAUDIN-1 (intestinal tight junction proteins) and PER1 (clock gene) were significantly higher in the intervention group. Also, in ZT13, there were different percentages of relative abundance in Firmicutes and Bacteroidetes phyla, being the first cited higher in the intervention group, but the second cited lower in the same group. Here, we observed different CFU/ml counting at ZT12 among the culture media, being the BA medium the lowest counting. This suggests a similar behavior observed of the Bacteroidetes phylum relative abundance in the control group. Anaerobic bacteria are also present in this phylum. Therefore, the result of Wang and colleagues seems to agree with our result. Furthermore, aiming to fully understand the mechanisms that underlie the diurnal rhythmicity of the gut microbiota, future studies should aim to evaluate specific molecules that underlie the physiology of the gastrointestinal tract.

Round 2

Reviewer 1 Report (New Reviewer)

I accept the revisions made to my comments. The changes improved clarity and representation of the work.

Author Response

We thank the reviewer for accepting our revisions.

Reviewer 2 Report (New Reviewer)

The authors have addressed my concerns in the writing. My one suggestion is that the authors may want to double check the error bars input for some of the points as they are different sizes above and below the point (example Feces BHI ZT0-12).   

Author Response

We thank the reviewer for accepting our revisions.

Reviewer 4 Report (New Reviewer)

Dear Author:

The revised manuscript is now detailed and the explanation of the important points are clear. Review comments  were addressed properly and the revised manuscript is acceptable for publication. 

Author Response

We thank the reviewer for accepting the revised version of the manuscript.

This manuscript is a resubmission of an earlier submission. The following is a list of the peer review reports and author responses from that submission.

Round 1

Author Response

Reviewer #1

The authors investigated diurnal oscillation of gut microbiota using cultivation techniques. As a communication paper, the preliminary results are presented. I find it interesting that the authors proposed the cultivation methods to understand the population of the microorganisms. I understand the overall idea that if this method is developed, it would lead to a less expensive and less time-consuming approach. However, the downside of using cultivation techniques including the inability to understand non-culturable microbes needs to be addressed properly. The idea is there but the investigation may not be well validated.

Reply: We thank the reviewer for the useful comments and suggestions, which we addressed below point-by-point.

  1. Even though the authors aim to use the cultivation method to study the metabolically active population, but the genera or species of the population were not identified. I understand that the selected culture media may be able to narrow down the group of bacteria that could grow. However, as this protocol is still in a developing stage, random selection of the colonies for species identification using the 16s rRNA gene may be a good practice upon protocol validation?

Reply: In this work, we aimed at demonstrating diurnal variation in the cultivable bacteria with high cell counts and cultivable in the gut microbiome. Given the relevance of the results, we decided to publish it as soon as possible, as a short communication, to provide the basis for future studies (which may combine culture and sequencing methods). To provide a protocol validation as the reviewer is suggesting, a future study could be performed for species identification, combining the culture of specific microbes and sequencing.

  1. What about the non-culturable microorganisms? I understand that the culture-dependent techniques allow us to see the metabolically active microbes, but the culture-independent techniques also allow us to see the whole population culturable or unculturable. How would the authors address the non-culturable microorganisms that may play a role in metabolisms as well?

            Reply: Regarding the non-culturable or difficult-to-grow microorganisms, we understand that molecular methods would allow for a deep understanding of the microbiome. As stated during the text, some microorganisms (e.g., Aspergillus fumigatus) would be interesting to be evaluated and we hope to be able to carry out a study that covers these gaps (Line 224). To address the non-culturable microorganisms that may play a role in metabolisms, a broad study should be performed comparing the results of culturomics and whole genome sequencing.

  1. In Line 57- new species cannot be taxonomically validated since biochemical and morphological analyzes cannot be performed, and microbial strains cannot be officially named. But if the "new species" are unculturable, can you taxonomically validate using your approach?

            Reply: No, but this question does not have much meaning here.  Our approach, as well explained, was aimed to address the diurnal variation of the whole microbiota. To address the difference between "culturable species" and "unculturable species", a broad study should be performed comparing the results of culturomics and whole genome sequencing. Then, for those "unculturable species", the proposal described by Liu et. al. (2022) may be applied, which indicates the way of using metagenomic data to bring uncultured microbes into cultivation.

  1. Following the previous question 2-3, do you think the media used in this work cover all groups of microorganisms?

            Reply: The non-specific culture media were used to allow the growth of large groups of microorganisms present in the intestinal microbiota, but these media certainly don’t cover them all. For that purpose, culturomics should be applied, as we have highlighted in the manuscript.

  1. Why these media were selected should be addressed clearly, including why the media favouring Streptococcus was used? And not other gut-dwelling bacteria?

            Reply: The choice of culture media was made considering the large groups of microorganisms present in the intestinal microbiota; and the presence of anaerobic microorganisms, especially Streptococcus, is quite high in this microenvironment. The number of media may be increased in future studies to cover more specific groups of gut microbes, especially for comparison with sequencing approaches.

  1. In Line 119- Petri plates with patterns indicative of contamination were excluded from analyses. Indicators of the contamination? From my understanding, we do not know for sure of which microbes would show up on the plate? Interestingly, if the colony with odd morphology could grow on the plate, does it suggest that the media selected in this work were not exclusively selective and how could we tell the difference between contaminants and the expected microorganisms then?

            Reply: We adopted the droplet plating method (Naghili et al., 2013), limiting the growth of bacteria in the areas where the drop with the inoculum was placed. The information was included to differentiate contaminants in the agar plate, such as the growth of microorganisms in the edge of the Petri dishes with the culture media, and the growth of microorganisms over the entire surface of the media.

  1. Do you think that it would be better if the data retrieved using this method is incorporated

with the metagenomic data? If yes, please demonstrate a case thoroughly of how the workflow would be.

            Reply: Yes, definitely. Our aim is indeed to develop further studies comparing data from both approaches. For this purpose, the same experimental design will be planned, but using media culture for more specific microbes, which will allow a detailed comparison with metagenomic data.

  1. More discussion on the culture-independent techniques regarding diurnal oscillation of gut

microbiota could be included. For example, interesting points that other researchers found that could be beneficial and prompt us to investigate more on diurnal oscillation of gut microbiota. And with the newly developed approach, could this information also be obtained? This point would make it very interesting as it suggests that your approach allow the discovery of the same useful information.

Reply: In fact, such discussion is outside the scope of this short communication. It will be quite useful, though, for the upcoming study we are planning, comparing culturable microorganisms vs metagenomic data, as explained in the previous answer.

Minor points

  1. The introduction should include the very important question of why we need to study diurnal oscillation of gut microbiota.

            Reply: We included this information, as requested.

  1. I find the introduction to be vague and pretty abstract. Specified examples and information

should be mentioned.

            Reply: As a short communication, there isn’t much space for detailed information in the introduction. So, we have chosen what would be suitable considering our main purpose with this work.

  1. The full name of all culture media should also be mentioned in Section 2.3

Reply: We modified figure 1 and placed the full name of the culture media above the abbreviations .

  1. Each panel in the figure could be enlarged and made clearer.

            Reply: We modified the figure, as requested.

  1. The collection of samples from faces is unclear.

            Reply: We added a sentence in section 2.2 with additional information (Lines 89 and 90).

Reviewer 2 Report

The title is non adequate. It must be reformulate

The informations in the introduction are very short, please develop it

Lines 80-82: Please Clarify the choice of these conditions

Lines 93-95: this part must be in the paragraph 2.1

Lines 102-103: Explain the choice of these dilutions

PBS: abreviation

Figure 1: Explain the conditions of growth of yeasts. Why you incubate at 26C the first day?????

All the media in the figure 1 must be write correctly and mention the company, country...

Explain the choice of the media BHI, SA, MS and BA

The section "Results" must be more explained

Verify the references in the text . example: Line 191, Line 219, Line 226.

The section references must be revised: Reference 2, 6, 8, 9, 15, 20, 21, 22 , 24

Author Response

Reviewer #2

Reply: We thank the reviewer for the useful comments and suggestions, which we addressed below point-by-point.

  1. The title is non adequate. It must be reformulated

            Reply: In our understanding, the title contemplates what we did in this study. Furthermore, from a chronobiological point of view, the most correct way to refer to these variations is as "diurnal variation", as we do not have time points covering other days in our study. So circadian variations, for example, wouldn't be correct either.

  1. The informations in the introduction are very short, please develop it

            Reply: As a short communication, there isn’t much space for detailed information in the Introduction. So, we have chosen what would be suitable considering our main purpose with this work.

  1. Lines 80-82: Please Clarify the choice of these conditions

            Reply: These conditions are the standard of a bioterium/vivarium or any housing of rats that is not intended to cause any intervention or stress on the animals. Since our goal was to assess the gut microbiota via microbial culture, we decided to keep the environment as controlled and "normal" as possible.

  1. Lines 93-95: this part must be in the paragraph 2.1

            Reply: We modified sections 2.1 and 2.2 to meet this need.

  1. Lines 102-103: Explain the choice of these dilutions

Reply: We chose the dilutions as determined in the previous studies from our laboratory. These dilutions allow for counting the CFUs without superposition of the colonies in the agar plates.

  1. PBS: abbreviation

            Reply: We put the full name of PBS (Phosphate buffered saline) in the first mention in the text.

  1. Figure 1: Explain the conditions of growth of yeasts. Why do you incubate at 26C the first day?????

Reply: The information was corrected in the manuscript. On the first day, we incubate AS plates at room temperature (25°C ±2°C), following the incubation at 37°C. As described by Hazen & Hazen (1987), C. albicans room-temperature grown cells were generally less sensitive to environmental perturbation and germinated more uniformly than cells grown at 37°C.

  1. All the media in figure 1 must be written correctly and mention the company, country…

            Reply: The information was included in Section 2.3.

  1. Explain the choice of the media BHI, SA, MS and BA

Reply: As previously answered, non-specific culture media in conditions to favor the growth of strict and facultative anaerobes would allow for the growth of a large group of cultivable bacteria. The medium for fungi was included to cultivate Candida spp. The Mitis Salivarius medium was used to allow for the Streptococci group growth. Therefore, it is expected that they vastly cover the expected populations in the gut microbiome. We added a sentence in section 2.3 to make it clearer.

  1. The section "Results" must be more explained

Reply: We consider that explaining the results in more detail means discussing them, as we already did in the Discussion.

  1. Verify the references in the text . example: Line 191, Line 219, Line 226.

Reply: We reviewed this session and removed references that did not match the information provided.

  1. The section references must be revised: Reference 2, 6, 8, 9, 15, 20, 21, 22, 24

            Reply: We reviewed this session and removed references that did not match the information provided.

Round 2

Reviewer 1 Report

Response 2: Should the last sentence be included in the manuscript as well?

Response 3: If no, then the sentence in Line 57 “new species cannot be taxonomically validated since biochemical and morphological analyzes cannot be performed, and microbial strains cannot be officially named.” may not be appropriate. Moreover, from my understanding, if the aim is to address the diurnal variation of the whole microbiota, you may want to include both "culturable species" and "unculturable species"? Especially when the “whole” microbiota is of interest?

Response 4: Should this be explained to the readers as well? To express that the authors have thoroughly thought of this point.

Response 5: Should this be explained to the readers as well? To express that the authors have thoroughly thought of this point.

Response 6: Can you observe the colony morphology using droplet plating method? The aim of this research is to develop a cultivation technique to investigate the microbiota, don’t the authors think it is also important to look at the difference in colony morphology?

Response 7: The authors should demonstrate the case thoroughly of how the workflow would be.

Response 8: I still think that a little bit more of a discussion as suggested is essential. From my understanding there are no word limits for the communication paper?  

Response minor 1: From my understanding there are no word limits for the communication paper?  

Reviewer 2 Report

The authors don't respond clearly to the comments .